# Urinary Biomarkers of Renal Injury KIM-1 and NGAL: Reference Intervals for Healthy Pediatric Population in Sri Lanka

**DOI:** 10.3390/children8080684

**Published:** 2021-08-09

**Authors:** P. Mangala C. S. De Silva, T. D. K. S. C. Gunasekara, S. D. Gunarathna, P. M. M. A. Sandamini, R. A. I. Pinipa, E. M. D. V. Ekanayake, W. A. K. G. Thakshila, S. S. Jayasinghe, E. P. S. Chandana, Nishad Jayasundara

**Affiliations:** 1Department of Zoology, Faculty of Science, University of Ruhuna, Matara 81000, Sri Lanka; chathura@zoo.ruh.ac.lk (P.M.C.S.D.S.); sameera.ac@live.com (T.D.K.S.C.G.); sakuntha.hope@gmail.com (S.D.G.); mihiri.a.sandamini@gmail.com (P.M.M.A.S.); isiniranawake@gmail.com (R.A.I.P.); emdvekanayake@gmail.com (E.M.D.V.E.); thakshilawanniarachchi@gmail.com (W.A.K.G.T.); 2Department of Biomedical Sciences, School of Medicine and Health Sciences, University of North Dakota, Grand Forks, ND 58203, USA; 3Department of Pharmacology, Faculty of Medicine, University of Ruhuna, Galle 80000, Sri Lanka; sudheerasj@yahoo.com; 4Department of Biosystems Technology, Faculty of Technology, University of Ruhuna, Matara 81000, Sri Lanka; epschandana@zoo.ruh.ac.lk; 5The Nicholas School of the Environment, Duke University, Durham, NC 27708, USA

**Keywords:** KIM-1, NGAL, children, CKDu, Sri Lanka, kidney biomarkers

## Abstract

Emerging renal biomarkers (e.g., kidney injury molecule-1 (KIM-1) and neutrophil gelatinase-associated lipocalin (NGAL)) are thought to be highly sensitive in diagnosing renal injury. However, global data on reference intervals for emerging biomarkers in younger populations are lacking. Here, we aimed to determine reference intervals for KIM-1 and NGAL across a pediatric population in Sri Lanka; a country significantly impacted by the emergence of chronic kidney disease of unexplained etiology (CKDu). Urine samples were collected from children (10–18 years) with no prior record of renal diseases from the dry climatic zone of Sri Lanka (N = 909). Urinary KIM-1 and NGAL concentrations were determined using the enzyme-linked immunosorbent assay (ELISA) and adjusted to urinary creatinine. Biomarker levels were stratified by age and gender, and reference intervals derived with quantile regression (2.5th, 50th, and 97.5th quantiles) were expressed at 95% CI. The range of median reference intervals for urinary KIM-1 and NGAL in children were 0.081–0.426 ng/mg Cr, 2.966–4.850 ng/mg Cr for males, and 0.0780–0.5076 ng/mg Cr, 2.0850–3.4960 ng/mg Cr for females, respectively. Renal biomarkers showed weak correlations with age, gender, ACR, and BMI. Our findings provide reference intervals to facilitate screening to detect early renal damage, especially in rural communities that are impacted by CKDu.

## 1. Introduction

Kidney disease is a significant global public health concern, impacting ~15% of the global population. Increased incidence of renal diseases and their progression to critical stages are confronting many nations across the globe. Socioeconomically disadvantaged countries with sub-optimal healthcare infrastructures are particularly burdened by kidney diseases [1,2,3]. This is highlighted in an emergence of chronic kidney disease of unknown etiology (CKDu) (also termed as chronic interstitial nephritis among agricultural communities) that is closely associated with agricultural lifestyle in several global hotspots mainly in Central America, and south-east Asia [4]. Although the causes of CKDu are unknown, a childhood-onset coupled to repeated acute kidney injury is suspected [5,6], highlighting the need for early-stage markers of kidney damage and dysfunction.

It is increasingly evident that repeated AKI can lead to the development of chronic kidney disease (CKD) and eventually end-stage renal disease (ESRD) through a progressive decline in renal function without apparent symptoms in the early stages [7]. The progression of renal diseases may be curtailed effectively if the disease or renal injury (e.g., AKI) is diagnosed earlier. Thus, a better estimation of early signs of kidney injury or damage is critical [8,9], especially in communities with a high prevalence of CKDu.

Considering the complex pathophysiology associated with chronic kidney disease, a single biomarker may not provide an early diagnosis. Multiple biomarkers may contribute to better diagnosis, prediction of renal outcomes, and elucidate associated pathophysiology while monitoring the effectiveness of therapeutic interventions [4]. Serum creatinine (sCr), estimated glomerular filtration rate (eGFR), and the albumin creatinine ratio (ACR), are commonly used standard renal biomarkers. Current clinical definitions for acute kidney injury (AKI) rely on serum creatinine sCr and urine output [10] while definitions of chronic kidney disease rely on eGFR and ACR [11]. However, detectable changes in these conventional markers may be delayed following renal injury [12], potentially leading to late diagnosis and underestimation of renal damage [13]. In addition, creatinine levels may vary significantly among individuals due to differences in muscle mass, age, gender, nutritional status, muscle metabolism, race, strenuous exercises, medications, and hydration status [14,15]. Thus, the prognostic value of these biomarkers in the early detection of renal impairment is challenging.

Highly specific and sensitive biomarkers are gaining increased attention in renal health research [16]. In contrast to the conventional markers, most of the emerging biomarkers lack adequate clinical validation, and detailed studies are required to determine the diagnostic and prognostic potential of these biomarkers in renal diseases. The United States Food and Drug administration (FDA-USA) has qualified several biomarkers including Kidney Injury Molecule-1 (KIM-1), Neutrophil Gelatinase-Associated Lipocalin (NGAL), N-acetyl-beta-D-glucosaminidase (NAG), Cystatin-C (CysC), clusterin (CLU), and osteopontin (OPN) as safety biomarker panels to assist in the detection of renal tubular injury in phase 1 trials [17]. Notably, multiple studies in diverse clinical settings and communities have demonstrated the role of KIM-1 and NGAL as potential markers of renal injury [4]. Furthermore, non-invasive approaches (i.e., urinary biomarkers) are critical in facilitating early diagnosis in these resource-limited communities.

Emerging biomarkers such as KIM-1 and NGAL may provide unique etiological insights into renal health in communities impacted by CKDu. However, reference intervals for these biomarkers have yet to be established for children in Sri Lanka, where CKDu is highly prevalent in certain agricultural regions. The aim of the present study was to define reference intervals of urinary KIM-1 and NGAL in a pediatric population in CKDu endemic and non-endemic regions in Sri Lanka and examine associations of these biomarkers with age, gender, and BMI, in order to strengthen the clinical screening system against kidney diseases.

## 2. Materials and Methods

### 2.1. Participants

Participants of both genders from selected government schools representing all the provinces in Sri Lanka were recruited for the study with their assent and, the consent of the parents. For the recruitment, the following inclusion criteria were adopted.

Aged between 10 and 18 years at the time of enrollment.Expressed consent of the children and the parents for participation, medical examination, donation of samples and long-term storage, and to produce records on medical history and current medications.Willingness to be contacted for future updates on medical status.Within normal BMI range (18.5–22.9 kg/m^2^) [18].

The following exclusion criteria were adopted to select biomarker data from the baseline biomarker database for the determination of reference intervals.

Known genetic disorders.Family history of chronic kidney disease.History or persistence of renal injury or disease kidney injury or disease, including renal stones, IGA Nephropathy, abnormal bladder, urinary infections, urinary reflux, and ureteral reimplantation.History or persistence of metabolic disorders, gastroesophageal reflux disease, gastrointestinal disorders.History or persistence of respiratory diseases including asthma, wheezing, and allergies.Hepatic diseases or impaired hepatic function detected in medical reports or a medical examination.BMI in underweight (<18.5 kg/m^2^), overweight (23–24.9 kg/m^2^) and obese (>25 kg/m^2^) ranges [18].Elevated ACR (>30 mg/g) in the urine samples collected within the present study.

All the participants were subjected to a medical examination with an investigation of medical history, current health status, and medications by medical professionals.

### 2.2. Sample and Data Management

First void early morning urine samples were obtained from each participant into a sterile container for the analysis. Samples were collected between 6–8 a.m. and brought to the collection points (schools) at room temperature and stored at 2–8 °C until centrifugation. The samples were centrifuged at 1000× *g* for 15 min at 4 °C and the supernatant was isolated. The supernatant was stored at −80 °C for the assessment of renal injury biomarkers.

An interviewer-administered structured questionnaire was used for the collection of demographic data, medical history, lifestyle habits, family history of diseases, current health status, and medications. The height and weight of the children were measured using a stadiometer.

### 2.3. Assessment of Renal Biomarkers

Renal injury biomarkers, KIM-1, and NGAL were assessed using Enzyme-Linked Immunosorbent Assay (ELISA) kits (Cusabio Technology LLC, Wuhan, China). Inter-assay precision and intra-assay precision values for the KIM-1 and NGAL ELISA kits were CV% < 10% and CV% < 8%, respectively [19]. Urine samples were analyzed for creatinine and microalbumin using an automated biochemistry analyzer (HumaStar 100; Human mbH, Wiesbaden, Germany).

### 2.4. Statistical Analysis

Baseline KIM-1 and NGAL concentrations in each urine sample were normalized to their creatinine content and expressed as adjusted concentrations. Creatinine-adjusted baseline KIM-1 ad NGAL concentrations in urine samples of the children were used for the determination of reference intervals for urinary KIM-1 and NGAL. The datasets were analyzed for outliers and Tukey’s method was adopted to remove outliers. The data were partitioned according to gender and stratified according to age prior to the analysis. The Shapiro–Wilk test was followed to determine the distribution pattern of creatinine-adjusted urinary biomarker concentrations [19,20]. The distribution of data deviated from a normal distribution towards a lognormal distribution, hence a nonparametric statistical approach was adopted. Kruskal–Wallis one-way analysis followed by Dunn’s multiple comparison test was used for comparison of creatinine-adjusted biomarker concentrations of samples of children in the same age range, collected from different locations [19,20]. Creatinine adjusted urinary biomarker concentrations of different children communities showed no significant difference, allowing them to be processed as a single group. The Mann–Whitney U test was used for the assessment of the effect of gender on biomarker concentration. Quantile regression analysis was adopted for the determination of reference intervals for biomarker concentrations. For each biomarker, the 2.5th, 50th, and 97.5th quantiles were calculated at a 90% CI [21]. Multiple linear regression analysis was followed to investigate the associations of biomarkers with age, gender, and BMI [19]. Statistical analysis was performed using Stata MP 16.0 (Stata Corp LLC, College Station, TX, USA) and IBM SPSS Statistics 26.0 (IBM INC., New York, NY, USA).

### 2.5. Ethical Considerations

The study was carried out under the approval of the ethics review committee of the Faculty of Medicine, University of Ruhuna, Matara, Sri Lanka (reference no: 2020.P.124 (20 November 2020)), in accordance with the declaration of Helsinki. Informed written consent of the parents along with the ascent of the children was obtained prior to participation in the study.

## 3. Results

A total of 909 children (male: 425; female:484) were selected for the study within the inclusion and exclusion criteria. The age of the participants was between 10 and 18 years, the mean age was 14.36 (SD:1.08) years, and the mean BMI was 18.35 (SD:3.70) kg/m^2^.

### Reference Intervals

The reference intervals for KIM-1 and NGAL (95%), including the 2.5th and 97.5th quantiles along with the median (50th quantile) and the related 90% confidence intervals, were determined by quantile regression, and are presented in Table 1 and Table 2 with stratification for gender and age. The quantile values reflect the biomarker reference value for the middle age within the respective age groups.

Overall, no significant difference in the KIM-1 levels was observed between males and females. However, statistically significant differences between age groups, particularly between the highest age group and lower age groups, were noted in both male and female cohorts (Figure 1).

Reference intervals (95%) and related 90% confidence intervals for creatinine-adjusted urinary NGAL are presented in Table 2.

Similar to urinary KIM-1, urinary NGAL showed no significant difference between males and females in the study groups. However, significant differences in NGAL levels among the age groups were noted, but only among females (Figure 2).

In the Spearman correlation analysis, KIM-1 showed a weak association with gender, age, and urinary NGAL level. In the gender-specific analysis, KIM-1 was weakly associated with age in both males and females. Further in females, KIM-1 was weakly associated with NGAL. However, no associations of KIM-1 were observed with ACR and BMI. Urinary NGAL showed significant correlations with age and ACR in spearman analysis without gender stratification. However, in the gender-specific analysis, NGAL showed significant correlations with age, ACR, BMI, and KIM-1 in females (Table 3).

## 4. Discussion

Here we report the first broad-scale study to determine reference intervals for KIM-1 and NGAL for a pediatric population in the Sri Lankan dry climatic zone, an area of the country significantly impacted by CKDu. The study focused on participants within the age range of 10–18 years from multiple geographical areas of the dry zone. However, creatinine-adjusted urinary biomarker levels showed no statistically significant difference between regions, allowing the data to be assessed as a single cohort. As per the guidelines of the Clinical and Laboratory Standard Institute (CLSI) guidelines (CLSI C28-A), the minimum sample size for the determination of reference intervals is referred to as 120 [22]. In our study, the total sample reached 909 resulting in higher precision of calculated reference intervals, a key strength of our study. Our data are limited to the age range of 10–18 years and due to practical limitations in participant recruitment. When considering age as a variable, for some age groups the sample size was comparatively low and is not consistent across all age groups. This remains a key limitation of our study that resulted from difficulty in recruiting participants from those age ranges. While studies are underway to recruit participants from underrepresented age groups, our current data provide a useful reference interval for kidney injury markers across a pediatric population.

Based on our analysis, we observed weak associations of KIM-1 and NGAL with age. Further, the present study reflected associations of KIM-1 with gender. However, no significant difference between male and female KIM-1 levels was noted. In addition, KIM-1 weakly correlated with NGAL, whereas NGAL weakly correlated with age, ACR, and BMI. In conventional diagnosis, ACR is used as a major criterion for characterizing renal abnormalities and stratification of CKD risk. As revealed in the study, the association of NGAL may be successfully incorporated with the utility of ACR through further validations.

According to our results, the range of median reference intervals (50th quantile) for urinary KIM-1 and NGAL were 0.081–0.426 ng/mg Cr, 2.966–4.850 ng/mg Cr for males, and 0.0780–0.5076 ng/mg Cr, 2.0850–3.4960 ng/mg Cr for females, respectively, for the age range 10–18 years. The values are consistent with the values observed in children of the same age range in several other countries. McWilliam et al. [21] reported median reference ranges for urinary KIM-1 and NGAL reference levels within the range 0.38–0.57 ng/mg Cr, and 6.3–63.3 ng/mg Cr respectively for Caucasian male and female children in UK and USA. Another study with 386 healthy children produced median concentrations of KIM-1 and NGAL levels without normalization to creatinine as 0.41 ng/mL (IQR 0.226–0.703 ng/mL) and 6.6 ng/mL (IQR 2.8–17 ng/mL), respectively, in urine samples [19]. In the present study, the median concentrations of KIM-1 and NGAL in urine samples were 0.11 ng/mL (IQR 0.003–0.285 ng/mL) and 2.859 (IQR 1.217–4.626 ng/mL), respectively, and are consistent with the above findings. However, the concentration of the biomarkers may vary depending on several factors including, water intake, time of sample collection, hydration status, and urine output. Hence, biomarker levels adjusted to creatinine may be more reliable.

In Sri Lanka, chronic kidney disease of uncertain etiology (CKDu) is prevalent in several regions and renal injury is suspected among pediatric populations in CKDu impacted regions [6]. Therefore, we examined urinary KIM-1 and NGAL levels of the participants residing in these areas in comparison to participants in areas where CKDu prevalence is not evident. Within this population, the median KIM-1 and NGAL levels for participants from disease-endemic areas were 0.129 ng/mg Cr (IQR 0.019–0.295 ng/mg Cr) and 2.830 ng/mg Cr (IQR 1.530–5.762 ng/mg Cr) respectively. For the participants from disease non-prevalent areas, the median KIM-1 and NGAL levels were 0.060 ng/mg Cr (IQR 0.002–0.210 ng/mg Cr) and 3.686 ng/mg Cr (IQR 1.818–6.478 ng/mg Cr), respectively. However, no statistically significant difference was observed between the biomarker levels of the disease prevalent and non-prevalent regions. In comparison, in Nicaragua, KIM-1 and NGAL levels above the healthy reference ranges indicating renal injury were observed in some participants from CKDu endemic regions [19]. Notably, as mentioned in our exclusion criteria, our renal biomarker data represent a pediatric population and we excluded any participants with ACR <30 mg/g from our analysis. Further, we did not include any participants with known renal disease or a family history of renal diseases. Relying on the reference values established here, further studies are underway to elucidate potential differences (as detected in our preliminary analysis and by Agampoidi et al. [6]) between children from CKDu-impacted and unimpacted regions in Sri Lanka.

Characterization of renal injury in AKI and CKD is still mostly based on conventional renal biomarkers such as albuminuria, sCr, ACR, and eGFR. The lack of validation studies, especially in rural under-resourced communities and among younger populations, has hindered the integration of these biomarkers into the clinical screening system for renal diseases. Thus, prompt studies for scientific validation with reference intervals for these biomarkers are critical. Given the asymptotic early stages and stealthy progression, early detection of renal damage or disease susceptibilities are highly important in the diagnosis management of chronic kidney diseases. Chronic kidney diseases, particularly CKDu, are likely mediated by several environmental factors (e.g., drinking contaminated water from agricultural activities) to which children and adolescents are more or less exposed. Thus, the onset of this disease may be apparent in early adulthood or at a young age. In Sri Lanka, early renal damage has been characterized in residential children in CKDu-affected areas, using conventional biomarkers [6]. Further, renal damage has been characterized in children in CKDu-affected regions in Nicaragua using novel biomarkers including KIM-1 and NGAL [23,24]. In Sri Lanka, KIM-1 and NGAL have been shown to be potential biomarkers in predicting renal damage in adult communities [25]. Thus, the development of reference standards for KIM-1 and NGAL in a pediatric population in our study will provide the foundation for integrating these biomarkers into the clinical screening system for effective non-invasive early diagnosis of renal diseases in rural communities.

## Figures and Tables

**Figure 1 children-08-00684-f001:**
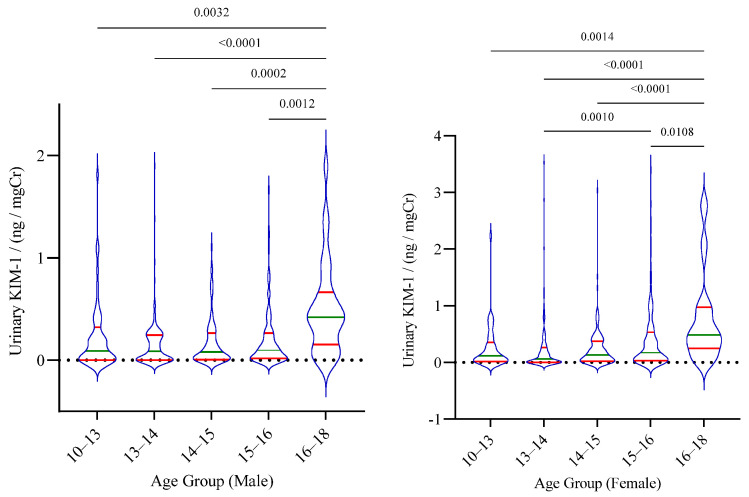
Distribution of urinary KIM–1 among the age groups of children. The median and inter-quartile range is shown for each group. Statistical significance is shown according to Kruskal–Wallis one-way analysis followed by Dunn’s multiple comparison test.

**Figure 2 children-08-00684-f002:**
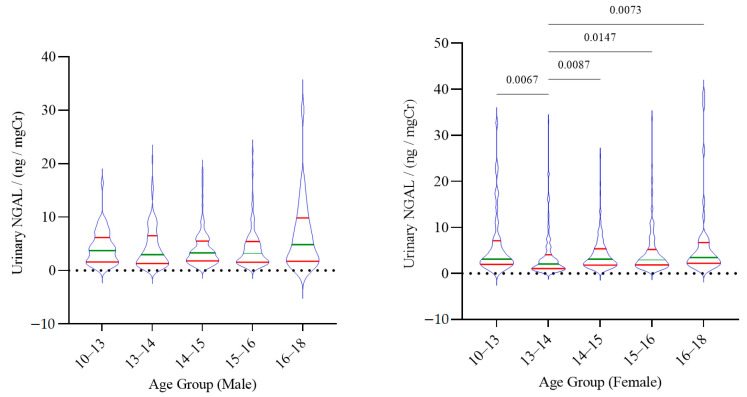
The distribution of urinary NGAL among the age groups of children. The median and inter-quartile range is shown for each group. Statistical significance is shown in adherence to Kruskal–Wallis one-way analysis followed by Dunn’s multiple comparison test.

**Table 1 children-08-00684-t001:** The reference intervals (RI) and confidence intervals (CI) for the creatinine-adjusted urinary KIM-1 levels for children. The ratio of the 90% confidence interval to the reference interval of the 97.5th quantile is given as the CI:RI ratio.

Age Group/Years	Urinary KIM-1 Concentration/(ng/mg Cr)			CI:RI Ratio
	2.5th Quantile (90% CI)	50th Quantile (90% CI)	97.5th Quantile (90% CI)	
**Male (N = 425)**				
10–13 (N = 50)	0.0008 (0.0003–0.0012)	0.0966 (0–0.1999)	1.1220 (0.3204–1.9236)	1.4289
13–14 (N = 123)	0.0005 0.0003–0.0007)	0.0939 (0.0290–0.1589)	0.7818 (0.3021–1.2616)	1.2272
14–15 (N = 112)	0.0005 (0.0003–0.0006)	0.0800 (0.0343–0.1258)	0.8813 (0.7267–1.0359)	0.3500
15–16 (N = 109)	0.0005 (0.0002–0.0007)	0.0984 (0.0558–0.1411)	1.1915 (0.8019–1.5811)	0.6539
16–18 (N = 31)	0.0009 (0–0.0367)	0.4262 (0.3534–0.4991)	1.8930 (1.3911–2.3949)	0.5303
**Female (N =484)**				
10–13 (N = 44)	0.0010 (0.0001–0.0019)	0.1188 (0–0.2625)	0.8253 (0–2.2014)	1.9990
13–14 (N = 127)	0.0006 (0.0005–0.0007)	0.0781 (0.0244–0.1319)	1.2904 (0.5197–2.0611)	1.1945
14–15 (N = 142)	0.0005 (0.0003–0.0006)	0.1395 (0.0868–0.1921)	0.9105 (0–1.9010)	2.0000
15–16 (N = 131)	0.0009 (0.0006–0.0013)	0.1737 (0.1188–0.2286)	2.3671 (1.4606–3.2735)	0.7659
16–18 (N = 40)	0.0008 (0–0.0792)	0.5076 (0.3751–0.6401)	2.8650 (2.6851–3.0450)	0.1256

**Table 2 children-08-00684-t002:** The reference intervals (RI) and confidence intervals (CI) for creatinine-adjusted urinary NGAL levels for children. The ratio of the 90% confidence interval to the reference interval of the 97.5th quantile is given as the CI:RI ratio.

Age Group/Years	Urinary NGAL Concentration/(ng/mg Cr)			CI:RI Ratio
	2.5th Quantile (90% CI)	50th Quantile (90% CI)	97.5th Quantile (90% CI)	
**Male (N = 425)**				
10–13 (N = 50)	0.4420 (0.0758–0.8082)	3.7475 (2.8171–4.6779)	9.6458 (5.3693–13.9223)	1.4433
13–14 (N = 123)	0.4240 (0.2728–0.5752)	2.9659 (2.1162–3.8155)	15.3648 (10.3116–20.4180)	1.3289
14–15 (N = 112)	0.6478 (0.4573–0.8383)	3.2992 (2.6910–3.9074)	15.9261 (10.0773–21.7750)	1.3672
15–16 (N = 109)	0.6895 (0.5671–0.8118)	3.2780 (2.5590–3.9970)	17.9484 (10.8294–25.0673)	1.3966
16–18 (N = 31)	0.4393 (0–1.0586)	4.8512 (1.9920–7.7104)	30.0482 (17.1954–42.9010)	1.4278
**Female (N = 484)**				
10–13 (N = 44)	0.8702 (0.2860–1.4544)	3.1455 (2.0366–4.2544)	23.4905 (11.1470–35.8340)	1.0509
13–14 (N = 127)	0.4189 (0.3221–0.5158)	2.0849 (1.5707–2.5992)	17.2057 (8.81067–25.6007)	0.9758
14–15 (N = 142)	0.7530 (0.4495–1.0566)	3.1281 (2.6457–3.6104)	13.5191 (6.8249–20.2134)	0.9903
15–16 (N = 131)	0.5727 (0.1730–0.9723)	2.9841 (2.6099–3.3584)	20.0642 (15.1797–24.9487)	0.4869
16–18 (N = 40)	1.1261 (1.0624–1.1898)	3.4958 (2.8220–4.1696)	38.9910 (33.3216–44.6603)	0.2908

**Table 3 children-08-00684-t003:** Associations of creatinine-adjusted urinary KIM-1 and NGAL levels with age, gender, BMI, and ACR as expressed in terms of the Spearman correlation analysis. Significant associations are shown in bold. The Spearman correlation coefficient is denoted by rs, where p represents the probability. BMI: body mass index; ACR: albumin creatinine ratio.

Variable	KIM-1		NGAL	
	rs	*p*	rs	*p*
**Unpartitioned (Male and Female)**				
Age (years)	**0.185**	**<0.0001**	**0.067**	**0.0004**
Gender	**0.109**	**0.001**	−0.034	0.308
BMI (kg/m^2^)	0.01	0.782	0.043	0.222
ACR (mg/g)	0.044	0.193	**0.094**	**0.005**
KIM-1 (ng/mg Cr)			**0.119**	**0.0004**
**Partitioned—Male**				
Age (years)	**0.137**	**0.005**	0.019	0.705
BMI (kg/m^2^)	−0.09	0.081	0.004	0.94
ACR (mg/g)	0.034	0.484	0.091	0.062
KIM-1 (ng/mg Cr)			0.048	0.323
**Partitioned—Female**				
Age (Years)	**0.215**	**<0.0001**	**0.116**	**0.012**
BMI (kg/m^2^)	0.053	0.272	**0.098**	**0.042**
ACR (mg/g)	0.028	0.543	**0.104**	**0.024**
KIM-1 (ng/mg Cr)			**0.192**	**<0.0001**

## Data Availability

According to the ethical libilities, we have not made the data available.

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
