# Peer review of "Urinary Biomarkers of Renal Injury KIM-1 and NGAL: Reference Intervals for Healthy Pediatric Population in Sri Lanka"

_children, 2021, doi:10.3390/children8080684_

Round 1

Reviewer 1 Report

The authors aimed to determine reference intervals for KIM‐1 and NGAL across a children population in Sri Lanka. A large, suitable cohort study was examined. The work has been conducted thoroughly. Below, please find a few comments on the manuscript.

  1. Due to some spelling errors, the manuscript should be carefully proofread, preferably by an English speaker (e.g., Page 2, line 47: citation is missed; page 2, line 77: ‘KIM-1 instead of ‘Kim 1’ – please unify it in the whole manuscript; Page 2, line 86: it should be ‘level’ instead of ‘lelve’).
  2. Page 3, line 109. Please explain ‘BMI in unhealthy range’ and define exactly what does ‘unhealthy’ mean? Underweight, overweight and obese or both? What type of references data was used to define those with abnormal BMI?
  3. Page 4, line 159: 425 males and 484 females does not equal 694 but 919. Please check these numbers throughout the manuscript.
  4. Page 4, line 161: please correct the units as fallow: ‘kg/m2
  5. Table 1 and Table 2. Please provide the exact number of subjects in each age group and explain the reason for combining children ages 10, 11, 12 and 13 into one group (the same for 16-18).
  6. Tables 1 and 2. There are the same numbers as upper limit in one group and the lower limit in the next group which can be confusing for the reader. Please try to change it so that it will be more transparent.
  7. There is no need to describe biomarkers levels (and 90% CI) to 7 decimal places. In case of KIM-1 it seems that 4 is fair enough (for NGAL 3 or 4). Likewise, in the case of p value and CI:RI ratio – 3 decimal places will suffice.
  8. Table 3. Based on the given Rs value (value of coefficient), one can infer quite differently than the authors report. The range between 0.00 to 0.19 means that the strength of correlation is very week, despite of the fact that difference is statistically significant. Therefore, this should be corrected in the text (Results and Discussion). Moreover, the conclusion should be drawn with cautions. I recommend omitting these results in the abstract.
  9. Please add units and explain abbreviations in Table 3.

Author Response

  1. Due to some spelling errors, the manuscript should be carefully proofread, preferably by an English speaker (e.g., Page 2, line 47: citation is missed; page 2, line 77: ‘KIM-1 instead of ‘Kim 1’ – please unify it in the whole manuscript; Page 2, line 86: it should be ‘level’ instead of ‘lelve’).

We have made the changes suggested by the reviewer and have proof read the manuscript carefully. 

- Abbreviation was made as KIM-1 and made consistent throughout the text

- Missing citation was added with the reference (LN: 49)

- Spelling mistakes and language errors throughout the text were fixed

  1. Page 3, line 109. Please explain ‘BMI in unhealthy range’ and define exactly what does ‘unhealthy’ mean? Underweight, overweight and obese or both? What type of references data was used to define those with abnormal BMI?

We have addressed these suggestions. BMI criteria with reference ranges were defined under inclusion and exclusion criteria.

  1. Page 4, line 159: 425 males and 484 females does not equal 694 but 919. Please check these numbers throughout the manuscript.

We have now address this and thank you for catching this mistake. Group size is 909 (425+484). The correction has been made in the text.

  1. Page 4, line 161: please correct the units as fallow: ‘kg/m2

We have corrected the units.

  1. Table 1 and Table 2. Please provide the exact number of subjects in each age group and explain the reason for combining children ages 10, 11, 12 and 13 into one group (the same for 16-18).

Thank you for this suggestion and we have provided an explanation for this approach. The size of sub groups (age groups) was included in Tables 1 and 2. The group size was determined based on practical limitations, and follows a similar publication (McWilliam et al., 2014).

Our initial intention was to recruit equal number of participants for each age category. However, participation of students in some age groups (10 - 13) and (16-18) were low resulting low number of students in 10-11, 11-12, 12-13, as well as 16-17 and 17-18 categories. Thus, in order to maintain adequate sample sizes for age groups between 10-13 and 16-18, we had to combine several groups.

We discussed this matter as a limitation of the study in the discussion (Ln:217-219)

  1. Tables 1 and 2. There are the same numbers as upper limit in one group and the lower limit in the next group which can be confusing for the reader. Please try to change it so that it will be more transparent.

We have now addressed this suggestion. Significant digits were adjusted and the issue is fixed now with the changes.

  1. There is no need to describe biomarkers levels (and 90% CI) to 7 decimal places. In case of KIM-1 it seems that 4 is fair enough (for NGAL 3 or 4). Likewise, in the case of p value and CI:RI ratio – 3 decimal places will suffice.

We have now addressed this suggestion. Values in the tables were adjusted to four decimal places.

  1. Table 3. Based on the given Rs value (value of coefficient), one can infer quite differently than the authors report. The range between 0.00 to 0.19 means that the strength of correlation is very week, despite of the fact that difference is statistically significant. Therefore, this should be corrected in the text (Results and Discussion). Moreover, the conclusion should be drawn with cautions. I recommend omitting these results in the abstract.

Thank you very much for the suggestion and we have now addressed it in the abstract and in results and discussion.

  1. Please add units and explain abbreviations in Table 3.

We have now addressed this suggestion. Units were added to the table and acronyms were defined in table caption.

Reviewer 2 Report

1.Please, explain: why you selected adolescent group? why not younger children? Whether this chronic kidney disease of uncertain origin appears in Sri Lanka  this age group? In my opinion bigger differences were in the group of younger children.

2.Results: please, explain:

-age of participants: it was 10-18 orf 11-18 (p.4, w. 160)?

-how large was your study group 694 or 909 (425+484)?

-how large were aged-related subgroups, whether they were comparable?

3. Use the same word abbreviation KIM-1 (not Kim-1)

Author Response

1.Please, explain: why you selected adolescent group? why not younger children? Whether this chronic kidney disease of uncertain origin appears in Sri Lanka this age group? In my opinion bigger differences were in the group of younger children.

In Sri Lanka the studies on renal injury among children is not extensively studied and this study serves as a starting point for a more comprehensive analysis. At this stage, our goal was not to examine difference between CKDu endemic and non-endemic regions, but to capture a wide population of adolescents from different parts of the country to establish reference intervals, especially for biomarkers such as KIM-1 and NGAL, that can potentially provide mechanistic insights into kidney injury.

One recent study illustrated that early renal injury (based on uACR and eGFR) maybe prevalent among residential children (6-11 years of age; N=2880), in CKDu affected areas in Sri Lanka (Agampodi et al., 2018). Accordingly, we would expect incidences of kidney injury to remain consistent to adolescent stages. However, prior to testing this hypothesis, we first wanted to examine the reference intervals for emerging biomarkers (KIM1 and NGAL) across a broad population.

Being consistent with the situation in Sri Lanka, a study in Nicaragua reported the prevalence of substantial renal damage among adolescents (age 7-17; N=210)(Leibler et al., 2021). Thus, in this study, we prioritized the school children in secondary school age (10-18 years of age) as they are in a relatively high risk next to the adults, particularly in the CKDu affected areas. Further, studies are underway to extend our analysis to include primary school children (below 10 years of age) in Sri Lanka.

We have now addressed this concern in lines 220-230.

2.Results: please, explain:

-age of participants: it was 10-18 or 11-18 (p.4, w. 160)?

Age of the participants is 10-18 years and it has been corrected in the text.

-how large was your study group 694 or 909 (425+484)?

Group size is 909 (425+484). The correction has been made in the text.

-how large were aged-related subgroups, whether they were comparable?

the size of sub groups (age groups) was included in the table 1 and table 2. The group size is acceptable as evidenced in similar publications(McWilliam et al., 2014).

  1. Use the same word abbreviation KIM-1 (not Kim-1)

Accepted. Abbreviation was made as KIM-1 and made consistent throughout the text.

Round 2

Reviewer 1 Report

The authors have addressed the points raised in my previous review.